# Antimicrobial Susceptibility, Virulence, and Genomic Features of a Hypervirulent Serotype K2, ST65 *Klebsiella pneumoniae* Causing Meningitis in Italy

**DOI:** 10.3390/antibiotics11020261

**Published:** 2022-02-17

**Authors:** Aurora Piazza, Matteo Perini, Carola Mauri, Francesco Comandatore, Elisa Meroni, Francesco Luzzaro, Luigi Principe

**Affiliations:** 1Clinical-Surgical, Diagnostic and Pediatric Sciences Department, Unit of Microbiology and Clinical Microbiology, University of Pavia, 27100 Pavia, Italy; aurora.piazza@unipv.it; 2Romeo and Enrica Invernizzi Pediatric Research Center, Department of Biomedical and Clinical Sciences L. Sacco, University of Milan, 20157 Milan, Italy; matteo.perini@unimi.it (M.P.); francesco.comandatore@unimi.it (F.C.); 3Microbiology and Virology Unit, A. Manzoni Hospital, 23900 Lecco, Italy; c.mauri@asst-lecco.it (C.M.); el.meroni@asst-lecco.it (E.M.); f.luzzaro@asst-lecco.it (F.L.)

**Keywords:** hypermucoviscous, hypervirulent, *Klebsiella pneumoniae*, ST65, virulence determinants, meningitis, invasive infection

## Abstract

The rise of a new hypervirulent variant of *Klebsiella pneumoniae* (hvKp) was recently reported, mainly linked to the ST23 lineage. The hvKp variants can cause severe infections, including hepatic abscesses, bacteremia, and meningitis, with a particularly disconcerting propensity to cause community-acquired, life-threatening infection among young and otherwise healthy individuals. The present study aimed to report the clinical characteristics of a hypermucoviscous *K. pneumoniae* strain isolated in Italy and sustaining recurrent meningitis in a patient of Peruvian origin. A further objective was to retrospectively investigate, by means of whole-genome sequencing (WGS) analysis, the genomic features of such an isolate. The hypermucoviscosity phenotype of the strain (sk205y205t) was determined using the string test. Genomic information was obtained by WGS (Illumina) and bioinformatic analysis. Strain sk205y205t was susceptible to most antibiotics, despite the presence of some resistance genes, including *bla*_SHV-11_, *bla*_SHV-67_, *fosA,* and *acrR*. The isolate belonged to ST65 and serotype K2, and exhibited several virulence factors related to the hvKp variant. Among these, were the siderophore genes *entB*, *irp2*, *iroN*, *iroB,* and *iucA*; the capsule-regulating genes *rmpA* and *rmpA2*; and the type 1 and 3 fimbriae *fimH27* and *mrkD*, respectively. A further operon, encoding the genotoxin colibactin (*clbA-Q*), was also identified. The virulence plasmids pK2044, pRJA166b, and pNDM. MAR were also detected. Phylogenetic investigation showed that this Italian strain is highly similar to a Chinese isolate, suggesting a hidden circulation of this hvKp ST65 K2 lineage.

## 1. Introduction

The rise of a new hypervirulent variant of *Klebsiella pneumoniae* (hvKp) was recently reported. The hvKp variant was first described in Taiwan, causing invasive pyogenic liver abscess [1,2], but can also be responsible for other invasive diseases, including abscesses at other body sites (e.g., eyes, brain, prostate, and kidney), necrotizing fasciitis, severe pneumonia with bacteremia, and meningitis [3]. Although hvKp infection appears to often occur in diabetic patients, a particularly disconcerting problem is its ability to cause community-acquired, life-threatening infection among young and healthy individuals [4].

The hvKp strains often present colonies on agar plates with the peculiar aspect of hypermucoviscosity, detectable semi-quantitatively by a positive ‘string test’; a widely used method for hvKp identification [5]. Usually, hvKp variants are susceptible to most antibiotics, except inherently resistant ampicillin, while several virulence factors have been identified [6]. The majority of hvKp show K1 or K2 capsular serotypes, even though non-K1/K2 serotype hvKp strains have also been reported and some K1/K2 strains are non-hvKp. Several sequence types (ST) have shown to be associated with hvKp, with ST23 being the most represented, related to the K1 serotype, followed by ST83 and ST65, associated with the K2 serotype [3].

An increasing number of studies are aiming at deciphering the virulence factor content and the specific feature of this *K. pneumoniae* variant. According to the genomic data, several virulence factors are tightly associated with the hvKp variant: the iron acquisition systems salmochelin (*iroBCDN*)/aerobactin (*iucABCDiutA*), the regulator of mucoid phenotype A gene (*rmpA/A2*), *magA*, *kfu*, *fimH*, *wabG*, *uge*, *allS*, a PhoPQ-activated integral membrane protein (*pagO*) and *entB* are the main virulence factors [7,8]. Usually, the virulence genes *iroBCDN*, *iucABCDiutA*, and *rmpA/A2* are carried on the virulence plasmid pK2044. Their presence was significantly associated with *K. pneumoniae* causing invasive human infection [9].

The present study aimed to report the clinical characteristics of a hypermucoviscous *K. pneumoniae* strain causing meningitis, and to retrospectively investigate, by means of whole-genome sequencing (WGS) analysis, the genomic features of such an isolate.

## 2. Case Description

On June 2021, a 75-year-old man of Peruvian nationality was admitted at the Medical Department of the ASST of Lecco (Italy). Based on clinical and laboratory findings, including fever, headache, sleepy state, and cerebrospinal fluid analysis, the patient was suspected of meningitis and empirically treated with ceftriaxone (4 g daily, bid), ampicillin (12 g daily, tid), and vancomycin (2 g daily, bid). Microbiology cultures showed both urinary tract and bloodstream infections caused by *K. pneumoniae*. Thus, treatment was changed to meropenem (6 g, tid) and linezolid (1200 mg bid). On August, following complete relief of signs and symptoms, the patient was discharged with diagnosis of decapitated meningitis and hyposodemia. One month later, on September 2021, the patient was readmitted at the ASST of Lecco. On admission, he had fever (38.8 °C), headache, heart rate of 70 beats per minute, blood pressure 119/83 mm Hg, 95% oxygen saturation, and white blood cell count of 12.9 × 10^9^/L. A tomographic scan of the brain and a chest X-ray were not suggestive of a central nervous system infection. However, based on laboratory and clinical data the presence of sepsis and meningitis was again suspected. Empiric therapy was implemented with ceftriaxone (4 g daily, bid) and ampicillin (12 g daily, tid). Lumbar puncture showed a xanthochromic turbid cerebrospinal fluid (CSF) with meningitis features: pleocytosis with a white blood cell count of 8758/mm^3^ (neutrophils, 97.4%). CSF also showed extreme hypoglycorrhachia (<1 mg/dL for a serum glucose level of 119 mg/dL) and high protein levels (1004 mg/dL). Given that a BioFire FilmArray^®^ Meningitis/Encephalitis panel (BioFire Diagnostics, Salt Lake City, UT, USA) was negative, meningitis caused by an uncommon bacterial etiologic agent was suspected. After a 10-h incubation period, an aerobic bottle from blood cultures performed at the emergency department was flagged positive by the instrument. Direct microscopic examination based on Gram staining evidenced the presence of Gram-negative rods. Empiric therapy was then changed to meropenem (6 g daily, tid), gentamicin (500 mg tid), and colistin (4.5 MU × 2, following a loading dose of 9 MU).

The patient had a Glasgow score of 7. Electroencephalography (EEG), brain nuclear magnetic resonance (NMR), and transesophageal echocardiography (TEE) were performed. EEG and NMR showed cerebral injury and hypodensity areas interpretable as vasculitic lesions, respectively, while TEE resulted negative. During the two weeks following the first CSF sample collection, blood cultures and a lumbar puncture were performed. Microbiology investigations resulted negative for these specimens. Nevertheless, this second CSF analysis again showed meningitis features: a white blood cell count of 80/mm^3^, a serum glucose level of 40 mg/dL, and 81 mg/dL protein levels.

Following clinical improvement, meropenem, gentamycin, and colistin were switched to ceftriaxone (4 g daily) and gentamycin (500 mg daily).

## 3. Laboratory Findings

### 3.1. Strain Identification and Antibiotic Susceptibility Test

Identification to the species level was performed by MALDI-TOF Mass Spectrometry (Vitek MS, bioMérieux, Marcy-l’Étoile, France), while susceptibility testing was routinely determined using a Vitek 2 system (bioMérieux) and Sensititre™ Gram Negative Plate (ThermoFisher, Waltham, MA, USA). Susceptibility results were interpreted according to current EUCAST criteria. Two sets of blood cultures and CSF were performed and sent to the laboratory. *K. pneumoniae* was isolated from both blood samples and CSF by culture in agar plates. The hypermucoviscosity phenotype of the two strains (sk205y205t and LC-1574/18) was determined using the string test. Hypermucoviscous *K. pneumoniae* (hmKp) was diagnosed by positive string test (>5 mm).

All isolates showed the same multi-susceptible profile: ertapenem (MIC, ≤0.12 mg/L, S), meropenem (MIC, ≤0.12 mg/L, S), imipenem (MIC, ≤0.5 mg/L, S), ceftazidime (MIC, ≤0.5 mg/L, S), ceftazidime/avibactam (MIC, ≤0.5 mg/L, S), ceftolozane/tazobactam (MIC, ≤0.5 mg/L, S), colistin (MIC, 0.5 mg/L, S), amikacin (MIC, ≤4 mg/L, S), gentamycin (MIC, ≤0.5 mg/L, S), ciprofloxacin (MIC, ≤0.06 mg/L, S), piperacillin/tazobactam (MIC, 4/4 mg/L, S), amoxicillin/clavulanic acid (MIC, 4/2 mg/L, S), tigecycline (MIC, 0.5 mg/L, S), and trimethoprim/sulfametoxazole (MIC, ≤1/19 mg/L, S). Detailed results are summarized in Table 1.

### 3.2. Genomic Characterization

The strains (isolated by culture in agar plates from CSF and positive blood culture) were subjected to whole-genome sequencing, as follows: genomic DNA was extracted with a QIAamp DNA minikit (Qiagen, Düsseldorf, Germany), following the manufacturer’s instructions, and sequenced on the Illumina Miseq platform with a 2 × 250 paired-end run, after a Nextera XT library preparation step (Illumina Inc., San Diego, CA, USA).

The reads quality was checked using FastqC software (https://www.bioinformatics.babraham.ac.uk/projects/fastqc/, accessed on 12 July 2020), and the low quality terminal residues were trimmed using Trimmomatic software (V0.36) (MAXINFO: targetLength = 50, strictness = 0.7) [10]. Genomic reads were assembled using SPAdes software (k-mer sizes: 21, 33, 55, 77, 99, 127 with “--careful” settings) [11]. The genome assembly was submitted to the European Nucleotide Archive (ENA), with the codes pending. The genomic distance among the study strain and the *K. pneumoniae* genomes retrieved from PATRIC database [12] was estimated using Mash software [13]. The 59 genomes most similar to the study strain were selected to perform the remaining analyses. The genomes were aligned against the reference *K. pneumoniae* genome ‘NTUH_K2044’ using progressive Mauve [14]. The core SNPs were named as described by Gona and colleagues [15]. The alignments of the core SNP sequences were subjected to phylogenetic analysis using RAxML software, with a 100 pseudo-bootstrap, after best model selection using ModelTest-NG [16].

The multilocus sequence type (MLST), the virulence gene content, the resistance genes, and the plasmid incompatibility groups of the strain were assessed using the Kleborate tool [17]. The variants of the genes *fimH* and *mr*kD, and the presence/absence of the gene *magA* were assessed using Blastn search (E-value threshold: 0.00001).

*K. pneumoniae* isolates obtained from CSF (sk205y205t) and blood cultures (LC-1574/18) were determined to be ST65 (*gapA*–*infB*–*mdh*–*pgi*–*phoE*–*rpoB*–*tonB*: 2–1–2–1–10–4–13) and capsular serotype K2.

Several virulence genes were detected, both specific and nonspecific for hvKp. At least two fimbriae-related genes were found: the *mrk* operon, which encodes type 3 fimbriae (*mrkABCDFHJ*), and the type 1 fimbriae *fim* operon (*fimH27*). The *wzi* gene, involved in the capsule attachment to the host cell surface and used for the prediction of capsular (K) antigen type, was found. Notably, the strain carried the *wzi-72* allele.

Regarding the iron acquisition systems, at least four systems were identified, including genes for aerobactin synthesis (*iutAiucABCD*), those for salmochelin biosynthesis (*iroBCDEN*), for yersiniabactin system (*fyuA-irp1-irp2-ybtSXQPAUET*), and enterobactin (*fepABCDG-entABCEF*). Genes encoding for aerobactin and salmochelin are known to be located on hvKp-specific, pLVPKlike virulence plasmids; while enterobactin and yersiniabactin coding genes are present in both hvKp and classical *K. pneumoniae* (cKp) strains.

Both isolates also contained the *rmpA* and *rmpA2* genes (regulators of the mucoid phenotype that increase capsule production), highly predictive of an hvKp strain. Virulence factors such as *wabG* (responsible for the biosynthesis of core lipopolysaccharide), and *uge* (uridine diphosphate galacturonate 4-epimerase), involved in capsule production were detected; thus, confirming the pivotal role of capsule-associated genes in the establishment of a hypervirulent behavior. Although less commons in hvKp, genes encoding colibactin (*clbA-Q*), a genotoxic metabolite, were also identified in sk205y205t and LC-1574/18 isolates.

It has already been established that serotype K2 isolates lack the *kfu* iron uptake gene and *allS*, a gene for allantoin metabolism; another gene required for hypermucoviscosity and virulence is *magA*, which appears to be restricted to K1 isolates. To confirm these features, the *Klebsiella* ferrous iron uptake system (*kfuA*, *kfuB,* and *kfuC* genes), *allS* and *magA* were not present in the two isolates. Furthermore, the strains also lacked the two-component system *kvgAS*, the microcin E495 and *peg-344* genes. Virulence factors and genomic features are shown in Table 2.

Plasmids harbored by the sk205y205t and LC-1574/18 strains were determined based on the assembled genome using the BLASTn program, which was able to match with plasmid sequences recorded in plasmid banks; as a result, plasmids pK2044, and pRJA166b were identified with 100% and 99.1% identity, respectively. A further identified plasmid was an IncHI1B plasmid, showing 99.3% identity with pNDM.MAR_JN420336.

In addition to virulence plasmids, hvKp strains have acquired other virulence genes through mobile genetic elements in the chromosome. The ICEKp10 encoding genes for colibactin synthesis had already been identified in the two isolates studied here.

The resistance mechanisms identified included the *bla*_SHV-11_, *bla*_SHV-67_, *oqxA*, and *oqxB* responsible for the resistance to beta-lactams; *fos*A that may be responsible for fosfomycin resistance; and *acrR,* which may confer resistance to fluoroquinolones. Nine different point substitutions were detected in the OmpK36 porin. Of these, N49S, L59V, L191S, F207W, D224E, L228V, and E232R have been previously associated to cephalosporins resistance; while A217S and N218H were associated to carbapenem resistance [18]. Three different point substitutions were found in the OmpK37 porin: I70M, I128M, and N230G, previously associated with carbapenem resistance [19]. Last, the following seven substitutions, observed on the AcrR sequence, P161R, G164A, F197I, F172S, R173G, L195V, and K201M had previously been associated with fluoroquinolone resistance [20].

A comparison of the sk205y205t isolate with other *K. pneumoniae* genomes deposited in the PATRIC database is shown in Figure 1. Notably, sk205y205t showed a major similarity with a ST65 strain collected in China. The Chinese isolate showed the same capsular serotype and virulence genes content, but a greater number of resistance genes, which also include *qnrS1*, *ermB*, *bla*_TEM-1d_, *bla*_CTX-M-14_, and *bla*_CTX-M-3_.

## 4. Discussion

Hypervirulent *K. pneumoniae* (hvKp) is an emerging and evolving pathotype. hvKp shows an increased ability to cause severe infections, including hepatic abscesses, central nervous system infections, and endophthalmitis, with a particularly disconcerting propensity to cause community-acquired, life-threatening infection among young and otherwise healthy individuals [21]. Incidence of infections due to hvKp has been steadily increasing over the last three decades, and although, for the moment, they are mainly susceptible to antibiotics, hvKp strains are becoming increasingly resistant to antimicrobials via acquisition of mobile elements carrying resistance determinants [2].

Several genetic factors that confer the hypervirulent phenotype have been identified. In particular, those involved in increased capsule production and iron uptake systems are established hvKp-specific virulence factors. In addition, this also includes high mucus, lipopolysaccharide, fimbriae-related (type I fimbriae and type III pili), and biofilm formation [22].

In this study we analyzed by WGS and bioinformatic analysis the genomic features of a hmKp isolate (sk205y205t) from a case of meningitis. We also studied a subsequent *K. pneumoniae* isolate, LC-1574/18, collected from a blood culture of the same patient, in order to assess if one strain was responsible for those infections. For both strains we identified multiple virulence genes, associated with hypervirulence, and few antibiotic resistance genes. In our case, the production of SHV-11 and SHV-67 determinants did not confer resistance to third generation cephalosporins; hence, showing a non-ESBL phenotype. The putative virulence plasmids harbored by the sk205y205t and LC-1574/18 isolates highly resembled the hvKp-specific pLVPKlike plasmid.

We also conducted a literature search of cases of meningitis sustained by hmKp/hvKp (Table 3 and Table 4) using the PubMed Database (only articles in English were taken into account) [23,24,25,26,27,28,29,30,31,32,33,34,35,36]. The following terms and keywords were used for searching: *Klebsiella* hypervirulent cerebrospinal fluid; *Klebsiella* hypermucoviscous meningitis; hypervirulent *Klebsiella pneumoniae* meningitis. Notably, the search revealed 16 cases of meningitis sustained by hmKp/hvKp. From literature data, it appears clear that some risk factors play an important role in hmKp/hvKp meningitis, such as diabetes mellitus (7/16), alcoholism (5/16), and chronic viral infections (i.e., HIV, HBV, HTLV; 4/16). Furthermore, in the majority of cases, no significant medical history was associated with the hvKp infection.

More than 50% of meningitis presented abscesses (mostly hepatic) as first localization. Strains causing meningitis mostly showed an antibiotic multi-susceptible phenotype. Death occurred in more than 50% of cases.

Our case represents the first description of meningitis sustained by hmKp/hvKp in Italy. In particular, the patient had a Peruvian origin. Notably, three cases (3/16) of meningitis from the scientific literature presented origins from Central-South America. The patient reported no medical history; hence, suggesting a community-acquired infection. No other specific risk factors were reported. Moreover, no abscess was revealed in our patient; thus, suggesting that meningitis occurred after sepsis (probably starting from urinary tract infection) sustained by hvKp. The hvKp strain showed an antibiotic multi-susceptible phenotype, as well as other strains causing meningitis from literature cases. Moreover, antimicrobial treatment improved the patient’s clinical condition, probably due to the absence of important risk factors or comorbidities.

The isolation of hmKp/hvKp strains represents an emergent cause of concern, often associated with invasive infections. This case highlights the severity of hypervirulent hypermucoviscous Kp with multiple and serious localizations in a few hours. The various virulence factors produced by this strain contributed to the invasive form of the infection. Moreover, the hypermucoviscous nature of the strain make very difficult a complete clinical resolution of the infection, despite the antibiotic multi-susceptible profiles. This aspect is often associated with recurrent infections, as occurred in our patient. This case also highlights the major contribution of the bacteriology laboratory, which allowed a rapid diagnosis by blood and CSF culture in association with a string test. This permitted the early detection of the hypermucoviscous phenotype and an optimal antibiotic treatment.

The string test method could be very useful for the rapid detection of the hypermucoviscous phenotype. However, clinical laboratories are unable to differentiate cKp from hvKp. The definition of biomarkers that can accurately predict hvKp strains (i.e., virulence factors associated to invasive infections) by WGS and the development of a more specific diagnostic test seem to be necessary for optimal patient care and epidemiologic surveillance of the dissemination of such strains.

## 5. Conclusions

The hypermucoviscous and hypervirulent phenotypes in *K. pneumoniae* represent a serious cause of concern, mostly causing severe invasive and recurrent infections. In particular, the meningitis sustained by hmKp/hvKp are associated with a mortality rate >50%. Fortunately, these strains are commonly associated with antibiotic multi-susceptible profiles, but resistant variants are emerging. The rapid detection of hmKp/hvKp phenotype, together with an optimal antibiotic treatment, could be of utmost importance for the clinical resolution of severe invasive infections sustained by these strains.

## Figures and Tables

**Figure 1 antibiotics-11-00261-f001:**
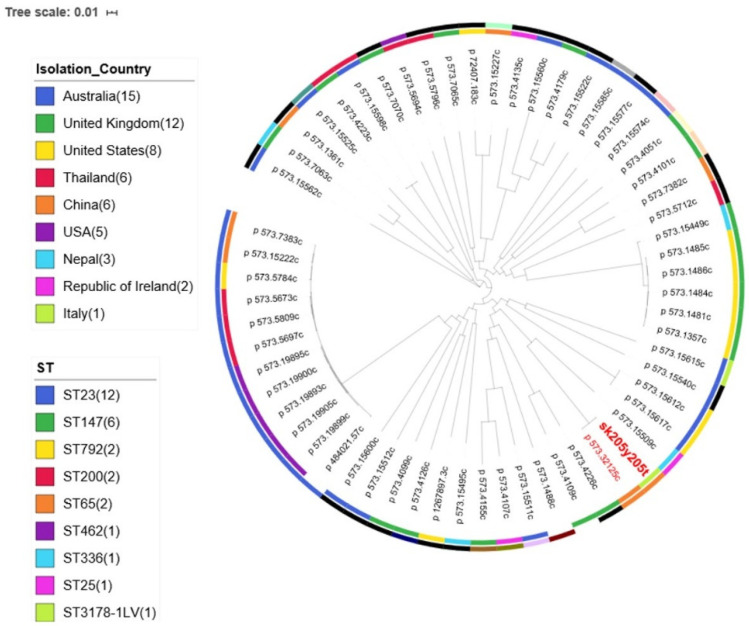
Maximum likelihood phylogenetic tree including the sk205y205t *K. pneumoniae* isolate (in red and bold type) and background strains retrieved from the PATRIC database. The geographic origin of all the strains is reported on the first level colored ring. The sequence type of the strains is indicated on the second colored ring.

**Table 1 antibiotics-11-00261-t001:** Susceptibility profiles of *K. pneumoniae* isolated from blood, cerebrospinal fluid, and urine. Minimum inhibitory concentrations (MICs) were interpreted using EUCAST criteria.

	Isolate from Blood	Isolate from CSF ^a^	Isolate from Urine
Antimicrobial Agent	MIC ^b^ (Interpretation)	MIC (Interpretation)	MIC (Interpretation)
Amoxicillin/clavulanic acid	≤2 (S)	≤2 (S)	≤2 (S)
Piperacillin/Tazobactam	≤4 (S)	≤4 (S)	≤4 (S)
Cefotaxime	≤0.25 (S)	≤0.25 (S)	≤0.25 (S)
Ceftazidime	≤0.5 (S)	≤0.5 (S)	≤0.5 (S)
Ertapenem	≤0.12 (S)	≤0.12 (S)	≤0.12 (S)
Meropenem	≤0.12 (S)	≤0.12 (S)	≤0.12 (S)
Imipenem	≤0.5 (S)	≤0.5 (S)	≤0.5 (S)
Amikacin	≤4 (S)	≤4 (S)	≤4 (S)
Gentamycin	≤0.5 (S)	≤0.5 (S)	≤0.5 (S)
Ciprofloxacin	≤0.06 (S)	≤0.06 (S)	≤0.06 (S)
Ceftazidime/Avibactam	≤0.5 (S)	≤0.5 (S)	≤0.5 (S)
Ceftolozane/Tazobactam	≤0.5 (S)	≤0.5 (S)	≤0.5 (S)
Colistin	0.5 (S)	0.5 (S)	0.5 (S)
Tigecycline	0.5 (S)	0.5 (S)	0.5 (S)
Trimethoprim/Sulfametoxazole	≤20 (S)	≤20 (S)	≤20 (S)

^a^ CSF: cerebrospinal fluid; ^b^ MIC values are in mg/L.

**Table 2 antibiotics-11-00261-t002:** Virulence factors and genomic features of the sk205y205t and LC-1574/18 *K. pneumoniae* strains.

Features of Virulence and Resistance	
HM phenotype	Positive
Capsule serotype	K2
Sequence type	ST65
**HM phenotype regulator genes**	
*rmpA*	+
*rmpA2*	+
**Siderophore systems**	
Enterobactin (*entABCEF*)	+
Aerobactin (*iucABCD*cluster)	+
Aerobactin receptor (*iutA*)	+
Yersiniabactin (*ybt* and *irp* complex)	+
Salmochelin (*iroBCD*)	+
Salmochelin receptor (*iroN*)	+
**Fimbrial genes**	
Type 3 fimbrial genes (*mrk* cluster)	+
Type 1 fimbrial genes (*fim* cluster)	+
**Genotoxin**	
Colibactin (*clbA* to *clbQ* cluster)	+
**Ferric uptake**	
*kfuABC* cluster	–
**Antibiotic-resistant genes**	
β-lactamases	*bla*_SHV-11_, *bla*_SHV-67_, *ompK*36, *ompK*37
Aminoglycoside resistance genes	–
Other resistance genes	*fosA*
Efflux pump associated genes	*acrR*, *oqxA*, *oqxB*

Notes: ‘+’ indicates the presence of the corresponding gene. ‘–’ indicates the absence of the corresponding gene. Abbreviation: HM, hypermucoviscosity.

**Table 3 antibiotics-11-00261-t003:** Clinical features related to cases of meningitis sustained by hmKp/hvKp, as extracted from the scientific literature.

Reference	Gender, Age, Nationality	Underlying Illness	Other Risk Factors	Medical History	Other Findings	Treatment	Outcome
This work	M, 75, Peruvian	No chronic illness	None	No significant medical history	Urinary tract infection, bloodstream infection	On admission: ceftriaxone and ampicillin then changed to meropenem, gentamycin and colistin. After laboratory findings and improvement of clinical conditions treatment was switched to ceftriaxone and gentamycin	Clinical improvement
Doud at al., 2009 [23]	M, 49, Afro-Carribean	Positive IgM and IgG for Dengue fever	None	No significant medical history	Multiple liver abscess	Meropenem, subsequently changed to ceftriaxone	Clinical resolution
Patel et al., 2013 [24]	F, 68, Guyanese	HTLV-1 positive, Alzheimer’s dementia, depression	None	Cervical cancer (treated 20 years before)	Multiloculated pyogenic liver abscess	Meropenem (2 g tid for 7 days), subsequently changed to ceftriaxone (2 g bid for 2 weeks) and then to amoxicillin/clavulanate (875 mg bid for 3 weeks)	Clinical resolution
Alsaedi et al., 2014 [25]	M, 62, Filipino	Chronic HBV infection	IgG2 deficiency	Treated pulmonary tuberculosis when he was 12, hypertension, mild chronic renal insufficiency, chronic obstructive lung disease	No foci of infection	Cefazolin (on admission) subsequently changed to ceftriaxone (>6 weeks therapy)	Clinical resolution
Melot et al., 2016 [26]	M, 55, Guadeloupean	Hypertension	Alcoholism	Benign prostate hyperplasia, hyperlipidemia, obstructive sleep apnea	Left mastoiditis, no liver abscess	Cefotaxime (4 g qid for 21 days)	Clinical resolution
Iwasakiet al., 2017 [27]	M, 72, Japanese	Hypertension, DM, cerebral hemorrhage without neurosurgical intervention, chronic pancreatitis	Alcoholism, heavy smoking (20 cigarettes daily)	5 days before admission, he was diagnosed with otitis media and underwent myringotomy (the discharge was not cultured)	No foci of infection	On admission: ampicillin, ceftriaxone, vancomycin; therefore changed with ceftriaxone alone	Clinical resolution
Khaertynov et al., 2017 [28]	M, 12-day-old, Russian	No illness	None	Full-term (40-week-gestation) neonate, born by cesarean delivery from a 24-year-old woman. The mother had no history of infections before delivery and no complications during pregnancy	No foci of infection	Ampicillin (200 mg/kg/day) and amikacin (10 mg/kg/day) on admission, then changed to meropenem (120 mg/kg/day) for 15 days and then to cefoperazone (100 mg/kg/day)	Cerebral edema and death
Maheswaranathan et al., 2018 [29]	F, 61, Chinese	No chronic illness	Impaired glucose tolerance without diagnostic criteria for DM	No significant medical history	Liver abscess	On admission: meropenem and intrathecal gentamycin. Meropenem was changed to cefepime and ciprofloxacin	She developed diabetes insipidus, uncal herniation and progressed to brain death
Hosoda et al., 2019 [30]	M, 71, Japanese	HBV and HTLV-1	Alcoholism	No significant medical history	Liver abscess. Chronic, but not disseminated, strongyloidiasis	Meropenem	Death due to cardiopulmonary arrest
Shi et al., 2019 [31]	M, 58, Chinese	No chronic illness	None	No significant medical history	Pulmonary abscess and bacteremia. Coinfection with *Cryptococcus neoformans*	On admission: imipenem-cilastatin, tigecycline, voriconazole then changed to meropenem combined with 5-fluorocytosine and fluconazole	Death due to respiratory and cardiac arrest caused by cerebral hernia
Rodrigues et al., 2020 [32]	Unknown	HIV infection, DM	Alcoholism	No recent hospital admissions	No foci of infection	Cefotaxime (2 g × 6 then increased to 3 g × 6) and ofloxacin. Treatment with cefotaxime was continued for 21 days.	Clinical resolution
Macleod et al., 2021 [33]	F, 60, Chinese	DM	None	No significant medical history	Liver and pulmonary abscess	On admission amoxicillin/clavulanate (1.2 g tid), empirical changed to cefotaxime (2 g qid) and acyclovir (10 mg/kg tid)	Death
Marinakis et al., 2021 [34]	F, 57, Filipino	DM untreated	None	No significant medical history	Abscess in liver segment VII and spleen	On admission: ceftriaxone, vancomycin, ampicillin/sulbactam, dexamethasone then de-escalated to ceftriaxone and subsequently changed to meropenem and tigecycline	Death in ICU due to MOF
	M, 50, Filipino	DM untreated	Alcoholism	No significant medical history	Abscess in both lungs and liver segment VII	On admission: ceftriaxone, vancomycin, dexamethasone	Death in ICU due to cerebral oedema
	M, 45, Filipino	DM untreated	Drug user	Deep neck abscess sustained by *K. pneumoniae,* successfully treated 9 months previously	Negative for liver abscess	On admission: meropenem, vancomycin, colistin, dexamethasone	Death in ICU due to MOF
Oh et al., 2021 [35]	F, 57, Unknown	No chronic illness-	None	No significant medical history	Liver abscess and cholecystitis	On admission: cefotaxime (2 g × 6), vancomycin (25 mg/kg every 24 h) followed by 18 mg/kg every 12 h) + dexamethasone (10 mg qid) then changed to ciprofloxacin (400 mg bid)	Death in ICU due to MOF
Troché et al., 2021 [36]	M, 54, Unknown	No chronic illness	None	No significant medical history	Liver abscess, multiple pulmonary nodules, endophthalmitis, hyperdense prostatic lesion and soft tissue abscess of both limbs	On admission: cefotaxime (200 mg/kg/die), acyclovir (30 mg/kg/die) then changed to ofloxacin and cefotaxime. After 6 weeks, cefotaxime was switched to trimethoprim-sulfamethoxazole with ofloxacin continuation	Clinical resolution

Abbreviations: DM, diabetes mellitus; MOF, multiple organ failure.

**Table 4 antibiotics-11-00261-t004:** Laboratory findings related to clinical cases of meningitis sustained by hmKp/hvKp.

Reference	Gender, Age, Nationality	Resistance Profile	Antibiotic Resistance Determinants	Virulence Factors	Capsule Serotype	Sequence Type	Note
This work	M, 75, Peruvian	-	*bla*_SHV-11_, *bla*_SHV-67_, *fos*A, *oqx*A, *oqx*B, *acr*R	*rmp*A, *rmp*A2, *mrk*ABCDFHJ, *fim*H27, *wzi-*72, *iut*A, *iuc*ABCD, *iro*BCDEN, *fyu*A, *irp*1, *irp*2, *ybt*SXQPAUET, *fep*ABCDG, *ent*ABCDFSE, *wab*G, *uge*, *clb*A-Q,	K2	ST65	String test positive
Doud et al., 2009 [23]	M, 49, Afro-Carribean	Ampicillin, ampicillin/sulbactam	-	(*mag*A negative)	K2	-	String test positive
Patel et al., 2013 [24]	F, 68, Guyanese	-	-	*rmp*A	-	-	String test positive
Alsaedi et al., 2014 [25]	M, 62, Filipino	Ampicillin, piperacillin	-	*rmp*A (*mag*A negative)	K1	-	String test positive
Melot et al., 2016 [26]	M, 55, Guadeloupean	Ampicillin, piperacillin	*bla* _SHV-1_	*rmp*A, *rmp*A2, *iro*BCDN, *iuc*ABCD, *iut*A, *kvg*AS, *mrk*ABCDFHIJ, *wzi*-2, *wzc*-2	K2	ST86	-
Iwasaki et al., 2017 [27]	M, 72, Japanese	Ampicillin, piperacillin	-	*rmp*A, *ter*A, *iro*N, *iuc*A	K54	ST29	String test positive
Khaertynov et al., 2017 [28]	M, 12-day-old, Russian	Ampicillin, amoxicillin/clavulanate, ceftazidime, cefotaxime, ceftriaxone, amikacin, gentamicin, ciprofloxacin	ESBL positive	*rmp*A	-	-	String test positive
Maheswaranathanet al., 2018 [29]	F, 61, Chinese	-	-	-	-	-	String test positive
Hosoda et al., 2019 [30]	M, 71, Japanese	-	-	*mag*A, *rmp*A, *iut*A, *fim*H, aerobactin, *iro*N	K1	ST23	String test positive
Shi et al., 2019 [31]	M, 58, Chinese	Ampicillin	-	-	-	-	String test positive
Rodrigues et al., 2020 [32]	Unknown	-	Isolate did not carry the intrinsic *bla*_SHV_ typical of Kp1, explaining the susceptibility to ampicillin. No acquired antimicrobial resistance gene was observed.	*clb*1, *ybt*12, *mrk* cluster, *pld*1	K2	ST66	-
Macleod et al., 2021 [33]	F, 60s, Chinese	Ampicillin	-	*rmp*A, *rmp*A2, *iuc*1, *iro*1, *clb/pks*, *ybt*	K1	Single-locus variant of ST23	String test positive
Marinakis et al., 2021 [34]	F, 57, Filipino	-	-	-	-	-	-
	M, 50, Filipino	Amoxicillin/clavulanate, cefotaxime, ceftazidime, cefepime, ceftriaxone	ESBL positive	-	-	-	-
	M, 45, Filipino	-	-	-	-	-	-
Oh et al., 2021 [35]	F, 57, Unknown	-	-	-	K1	-	String test positive
Troché et al., 2021 [36]	M, 54, Unknown	-	-	-	-	-	String test positive

## Data Availability

The genome assemblies of the sequenced strain have been submitted to European Nucleotide Archive (ENA), ID code pending.

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
