# Peer review of "Antimicrobial Susceptibility, Virulence, and Genomic Features of a Hypervirulent Serotype K2, ST65 Klebsiella pneumoniae Causing Meningitis in Italy"

_antibiotics, 2022, doi:10.3390/antibiotics11020261_

Round 1
Reviewer 1 Report
Review report:
The authors present a case of K.pneumoniae meningitis caused by a hypervirulent strain. The case is well documented, and the in-depth analysis of virulence factors is impressive. However, there are some issues that I consider necessary to be addressed in order to increase the value of the manuscript:
Major revision:
Issue 1: The sections of the manuscript should be organized different: There should not be a results section in a case report, but rather the paragraph should be named Case report. The materials and methods section are also not expected in a case report. This information should rather be inserted directly in the text of the manuscript.
Minor revisions:
Issue 2: English editing is recommended (ex. Was suspected of instead of for, line 78).
Issue 3: Please give information about the manufacturer, city, country for Filmarray line 95. This is also applicable for other devices used described in the methods subsection.
Issue 4: Cerebral pain (line 105) should be reformulated – I suggest cerebral injury?
Issue 5: I do not understand the meaning of the paragraph between line 107 and line 109. What do you mean? – please rephrase.
Issue 6: Please use italic font for K.pneumoniae throughout the whole manuscript (line 115-116).
Issue 7: Please insert references for the information provided in Discussion section lines 238-251.
Issue 8: Please give more details of the methodology used for the literature search (MeSh terms, Keywords used etc) and insert the information in the Discussion section line 263-264.
Issue 9: English grammar problem “there would not seem to be”? – line 269.
Author Response
Reviewer 1
Major revision:
Issue 1: The sections of the manuscript should be organized different: There should not be a results section in a case report, but rather the paragraph should be named Case report. The materials and methods section are also not expected in a case report. This information should rather be inserted directly in the text of the manuscript.
The text has been reorganized, as suggested. M&M have been incorporated in the section “Laboratory findings”.
Minor revisions:
Issue 2: English editing is recommended (ex. Was suspected of instead of for, line 78).
Thanks. We corrected the sentence. English language has been revised where appropriate.
Issue 3: Please give information about the manufacturer, city, country for Filmarray line 95. This is also applicable for other devices used described in the methods subsection.
Information have been added, where appropriate.
Issue 4: Cerebral pain (line 105) should be reformulated – I suggest cerebral injury?
Thank you. The term has been reformulated as suggested.
Issue 5: I do not understand the meaning of the paragraph between line 107 and line 109. What do you mean? – please rephrase.
We rearranged the sentence.
Issue 6: Please use italic font for K. pneumoniae throughout the whole manuscript (line 115-116).
We now used italic font, where appropriate, in all over the text.
Issue 7: Please insert references for the information provided in Discussion section lines 238-251.
We added two more references about the sentences reported in that part of the discussion, as suggested.
Issue 8: Please give more details of the methodology used for the literature search (MeSh terms, Keywords used etc) and insert the information in the Discussion section line 263-264.
Keywords have been added.
Issue 9: English grammar problem “there would not seem to be”? – line 269.
Thanks for the correction, we rewrite the sentence.

Reviewer 2 Report
I would only want to recommend that the antimicrobial susceptibility pattern should also be mentioned in the title since the manuscript is submitted to a journal including articles on antibiotics. The title only refers to its virulence. Maybe add something like: "antimicrobial susceptible strain"
Author Response
Reviewer 2
I would only want to recommend that the antimicrobial susceptibility pattern should also be mentioned in the title since the manuscript is submitted to a journal including articles on antibiotics. The title only refers to its virulence. Maybe add something like: "antimicrobial susceptible strain".
The title has been changed.

Reviewer 3 Report
Piazza et al. present an atypical meningitis clinical case, the subsequent isolation of its causative agent, followed by the phenotypic and genetic characterization of the bacterial isolate – a hypermucoviscous/hypervirulent (hv) Klebsiella pneumoniae strain. The authors also conducted a phylogenetic studies that suggest a hidden circulation of the presented hv K. pneumoniae K2 lineage which has the potential to reach global dimensions.
I have read the manuscript with interest and I have some minor suggestions:
Ln 81, Ln 115 – K. pneumoniae in italics
Ln 100 – “bacilli” may suggest a taxonomic class of (Gram-positive) bacteria. I would replace bacilli with rods for a better clarity
Ln 180 – The strain is described as entABCFSE positive in the text Ln 147-150?
Ln 194 – ompK in italics
Ln 211 – I would replace “synthetic” with “synthesis” for clarification
Ln 251 – pili
Ln 254 – I would change further with subsequent
Ln 266 – I would suggest that the authors remove “Asian origin” because such kind of statement might be due to literature bias.
Table 3 – spell out Cryptococcus neoformans
Ln 78 (new line numbers after tables) - Supplementary materials. The authors deposited a supplementary table which is similar to Table 3.
Ln 131 and 155 - K. pneumoniae in italics
Ln 152 – meningitis is not in italics
Author Response
Reviewer 3
Piazza et al. present an atypical meningitis clinical case, the subsequent isolation of its causative agent, followed by the phenotypic and genetic characterization of the bacterial isolate – a hypermucoviscous/hypervirulent (hv) Klebsiella pneumoniae strain. The authors also conducted a phylogenetic studies that suggest a hidden circulation of the presented hv K. pneumoniae K2 lineage which has the potential to reach global dimensions.
I have read the manuscript with interest and I have some minor suggestions:
Ln 81, Ln 115 – K. pneumoniae in italics
Italic font has been added where appropriate.
Ln 100 – “bacilli” may suggest a taxonomic class of (Gram-positive) bacteria. I would replace bacilli with rods for a better clarity.
Thanks, we have changed, as suggested.
Ln 180 – The strain is described as entABCFSE positive in the text Ln 147-150?
The strain was positive for entABCEF. This was adjusted also in Table 2.
Ln 194 – ompK in italics
Italic font has been added where appropriate.
Ln 211 – I would replace “synthetic” with “synthesis” for clarification
We have changed, as suggested.
Ln 251 – pili
We have changed as suggested.
Ln 254 – I would change further with subsequent
We have changed, as suggested.
Ln 266 – I would suggest that the authors remove “Asian origin” because such kind of statement might be due to literature bias.
“Asian origin” has been removed, as suggested.
Table 3 – spell out Cryptococcus neoformans
Done.
Ln 78 (new line numbers after tables) - Supplementary materials. The authors deposited a supplementary table which is similar to Table 3.
We have some problems with the “Antibiotics” format when big tables were added in horizontal pages. So big tables were submitted separately, as supplementary material, according to the Editorial Office. Tables (and line and page numbers) will be ok in the final version of the manuscript (without supplementary materials).
Ln 131 and 155 - K. pneumoniae in italics
Italic font has been added where appropriate.
Ln 152 – meningitis is not in italics
“Meningitis” is not in italics in all over the text.

Reviewer 4 Report
The manuscript is well organized and the scientific message of the research is interesting. Despite characterizing a susceptible isolate, as the authors have identified genomic features associated with resistance mechanisms, this case report could still be considered as under the scope of Antibiotics.
To my understanding, the study lacks further analysis of the listed substitutions (L. 213 – 220) on the genomic features accountable for resistance mechanisms. There are currently tools, like SIFT, PolyPhen-2, or PROVEAN, that predict whether an amino acid substitution is likely to affect protein structure and function based on sequence homology. Usage of such tools would improve the characterization of this hypermucoviscous K. pneumoniae strain causing meningitis.
The description of the Genome assembly should be revised to include more details, for instance regarding the quality threshold considered for initial trimming and assembly parameters in SPAdes (if that is the case, at least mention default parameters).
Moreover, the manuscript requires proofreading and language corrections. Just two examples, in L. 160, I believe the authors wished to say “Although” instead of “Althought”, and in L. 166, “features” should replace “festures”.
Minor comments:
- Throughout the manuscript, the usage of italics for species should be ensured (L. 81 and 115) and resistance genes should be correctly formatted with subscript (L. 235), as the authors have done in other mentions.
- Table 1 could be simplified, including the interpretation under parenthesis only in one column per isolate.
- Table 3, for the column “Other risk factors”, “none” should be used instead of “no one”. Here, for Maheswaranathan et al., 2018, the authors should correct “diasgnostic” to “diagnostic”.
- After Table 4, the numbering of lines and pages was restarted.
- The authors mention that susceptibility testing was determined by the Vitek 2 system and Sensititre Gram Negative Plate. Mentioning of following EUCAST guidelines for the MIC determination and interpretation should be made in the Materials and Methods section.
- In L.40 of section 4. Materials and Methods, the authors should verify if they indeed meant they have cultured the strains from liquor, and not “liquid”
- The References section should be revised, to ensure it follows journal's guidelines.
Author Response
Reviewer 4
The manuscript is well organized and the scientific message of the research is interesting. Despite characterizing a susceptible isolate, as the authors have identified genomic features associated with resistance mechanisms, this case report could still be considered as under the scope of Antibiotics.
Thank you for the nice comment.
To my understanding, the study lacks further analysis of the listed substitutions (L. 213 – 220) on the genomic features accountable for resistance mechanisms. There are currently tools, like SIFT, PolyPhen-2, or PROVEAN, that predict whether an amino acid substitution is likely to affect protein structure and function based on sequence homology. Usage of such tools would improve the characterization of this hypermucoviscous K. pneumoniae strain causing meningitis.
Thank you for this comment. The substitutions listed in the manuscript aimed to better characterise the strain: even though our strain is sensitive, those point mutations were previously detected in resistance strains. As suggested, some details on this point were lacking in this section of the manuscript. For this reason, we now specified which resistances are associated to such substitutions and we added to the references to the works where such substitutions were found in resistance strains.
The description of the Genome assembly should be revised to include more details, for instance regarding the quality threshold considered for initial trimming and assembly parameters in SPAdes (if that is the case, at least mention default parameters).
The requested details were added to the manuscript.
Moreover, the manuscript requires proofreading and language corrections. Just two examples, in L. 160, I believe the authors wished to say “Although” instead of “Althought”, and in L. 166, “features” should replace “festures”.
Thank you. English language has been revised where appropriate.
Minor comments:
Throughout the manuscript, the usage of italics for species should be ensured (L. 81 and 115) and resistance genes should be correctly formatted with subscript (L. 235), as the authors have done in other mentions.
Thanks for the suggestions. We revised all the text and corrected all the genus and species when left not in italics. Alike, we have standardized the way we reported the genes in the text.
Table 1 could be simplified, including the interpretation under parenthesis only in one column per isolate.
Thanks for the comment. We re-formatted the table according to your suggestion.
Table 3, for the column “Other risk factors”, “none” should be used instead of “no one”. Here, for Maheswaranathan et al., 2018, the authors should correct “diasgnostic” to “diagnostic”.
We replaced “no one” with the suggested “none” in the “Other risk factors” column and corrected the word “diagnostic”.
After Table 4, the numbering of lines and pages was restarted.
We have some problems with the “Antibiotics” format when big tables were added in horizontal pages. We have already contacted the Editorial Office when submitted the first version of this article. The page numbers will be ok in the final version of the manuscript.
The authors mention that susceptibility testing was determined by the Vitek 2 system and Sensititre Gram Negative Plate. Mentioning of following EUCAST guidelines for the MIC determination and interpretation should be made in the Materials and Methods section.
We add this information, as suggested.
In L.40 of section 4. Materials and Methods, the authors should verify if they indeed meant they have cultured the strains from liquor, and not “liquid”.
Samples from positive blood cultures and from cerebrospinal fluid were cultured on agar plates. We specified this in the new section “Laboratory findings”
The References section should be revised, to ensure it follows journal's guidelines.
References have been adjusted, as suggested.
